# The COVID-19 Vaccination Still Matters: Omicron Variant Is a Final Wake-Up Call for the Rich to Help the Poor

**DOI:** 10.3390/vaccines10071070

**Published:** 2022-07-03

**Authors:** Piotr Rzymski, Agnieszka Szuster-Ciesielska

**Affiliations:** 1Department of Environmental Medicine, Poznań University of Medical Sciences, 60-806 Poznań, Poland; 2Department of Virology and Immunology, Institute of Biological Sciences, Maria Curie-Skłodowska University, 20-033 Lublin, Poland

**Keywords:** pandemic, viral evolution, disease severity, SARS-CoV-2, vaccine inequity

## Abstract

By June 2022, COVID-19 vaccine coverage in low-income countries remained low, while the emergence of the highly-transmissible but less clinically-severe Omicron lineage of SARS-CoV-2 has led to the assumption expressed outside the academic realm that Omicron may offer a natural solution to the pandemic. The present paper argues that this assumption is based on the false premise that this variant could be the final evolutionary step of SARS-CoV-2. There remains a risk of the emergence of novel viral subvariants and recombinants, and entirely novel lineages, the clinical consequences of which are hard to predict. This is particularly important for regions with a high share of immunocompromised individuals, such as those living with HIV/AIDS, in whom SARS-CoV-2 can persist for months and undergo selection pressure. The vaccination of the least-vaccinated regions should remain the integral strategy to control viral evolution and its potential global consequences in developed countries, some of which have decided to ease sanitary and testing measures in response to the rise and dominance of the Omicron variant. We argue that low-income countries require help in improving COVID-19 vaccine availability, decreasing vaccine hesitancy, and increasing the understanding of long-term vaccination goals during the circulation of a viral variant that causes milder disease.

## 1. Introduction

When COVID-19 vaccines became available, the world gasped with high hopes of putting SARS-CoV-2 under better control. After some initial difficulties with the availability of doses, vaccine hesitancy was the only obstacle to COVID-19 vaccination campaigns in developed countries. At a time when booster doses were offered in various regions such as Israel, the USA, and the European Union, over a billion individuals of the African population remained unvaccinated, primarily due to a lack of vaccine availability. Initiatives that aim to ensure equitable access to COVID-19 vaccines in low-to-middle-income countries did not receive enough support. By the end of 2021, COVAX had delivered only 40% of the two billion doses scheduled for 2021 [1].

The primary reasons behind the weakening of the COVAX and global vaccine equity efforts lie in vaccine nationalism, summarized by the WHO’s Director-General as a “handful of rich countries gobbling up the anticipated supply as manufacturers sell to the highest bidder, while the rest of the world scrambles for the scraps” [2]. The direct response from wealthy nations has also been subdued, with an excess of unused doses being destroyed or sold to other developed regions.

Researchers have warned that vaccine inequity not only reflects a moral crisis but is increasing the odds of the emergence of novel, problematic SARS-CoV-2 variants [3]. The mutation frequency negatively correlates with the percentage of fully vaccinated individuals in a population, with the highest frequency found for regions with vaccination rates below 10–20% [4]. At the end of June 2022, only 18.5% of the African population had completed a primary vaccine protocol (Figure 1), compared to 73.2% in the European Union and 66.9% in the USA. Generally, only 16% of people in low-income countries have received at least one dose [5]. At the same time, high-income countries have increasingly authorized and recommended a second booster dose to maintain high levels of protection during upcoming waves of infections caused by the emerging Omicron sublineages [6,7].

The Omicron variant, which became dominant in various world regions by the end of 2021/beginning of 2022, has been evidenced through in vitro, in vivo, and epidemiological studies, as well as observations from clinical trials to cause milder infections [9,10,11,12]. On the other hand, there is also evidence that pre-existing immunity, as a result of infection and/or vaccination, plays a role in the higher frequency of milder outcomes [13]. As demonstrated, unvaccinated adults, adolescents, and children have higher odds of being hospitalized due to infection with Omicron [14,15,16]. Despite the lesser clinical severity of Omicron infections compared to those caused by other SARS-CoV-2 variants, the rising rate of hospitalizations due to this variant was in some countries, e.g., in the United States or Germany, either comparable to or even higher than during the Delta wave [17].

The observation of a milder clinical course of Omicron infection [18] may lead to the assumption that high vaccine coverage is no longer needed, since the Omicron provides a final solution to the pandemic [19,20]. Here, we argue that this assumption, which may also add to the loss of interest in vaccination in regions such as Africa, is built on the false premise that the Omicron variant is the final evolutionary step of SARS-CoV-2. We warn against the potential rise of novel subvariants, recombinants, and lineages, and the reemergence of previously-dominant variants. The vaccination of the least-vaccinated regions should remain the integral strategy to decrease the evolution of SARS-CoV-2 and the consequences it may have on pandemic dynamics. These regions should follow the recommendations on primary vaccination course, and the administration of additional and booster doses in specific demographic groups, which are issued in high-income areas, e.g., the United States and the European Union. There is no indication that low-income countries should have vaccine recommendations shaped differently to high-income regions, because COVID-19 is a worldwide issue and should be treated equally regardless of one’s origin or ethnicity. This is particularly important if one considers that due to the dominance of the Omicron variant, some developed countries (e.g., in Europe) have started to ease various sanitary measures such as the use of face masks in public places and limiting testing for SARS-CoV-2 infections.

## 2. The Rise of Omicron Is Not Equal to the Game over to COVID-19 Pandemic

The Omicron variant (B.1.1.529 lineage) was first identified in Botswana when the percentage of people living with HIV exceeded the rate of those who had completed an initial protocol of COVID-19 vaccination. With over 50 mutations accumulated in the genome, approximately 30 concerning spike protein and 10 in the receptor-binding domain [21], it is likely that B.1.1.529 emerged in an immunocompromised patient or as a result of cross-infection in a group of such individuals. It is well established that SARS-CoV-2 can persist for months in patients with advanced HIV disease due to the diminished immune response, which at the same time is sufficient to pressure the selection of extensive immune escape [22,23]. Africa has the highest population of HIV-positive people of all the world regions (predominantly in the Sub-Saharan region) [24], the majority of them still awaiting the COVID-19 vaccine. Since the initial appearance of Omicron in Africa, its various sub-variants have been identified, as well as recombinants not only of these sub-variants but also of Omicron and Delta [25]. Some of them are characterized by a transmission higher than the initial B.1.1.529. While there is no evidence for the greater pathogenicity of these sublines, it shows the potential of SARS-CoV-2 variability under insufficient control. Again, the rise of novel subvariants, as well as recombinants, is more likely in immunocompromised individuals because the persistence of SARS-CoV-2 increases the odds of harboring a co-infection with another viral variant. There are already documented cases of Omicron sub-variants and Omicron/Delta co-infections in patients with a weakened immune system [26]. This is particularly concerning if one considers that during the COVID-19 pandemic, the measures to control the burden of HIV/AIDS in the African region have been significantly subdued [27,28]. As Omicron has evolved, its increasing adaptation to evade infection-acquired or vaccine-induced immunity and lowering the effectiveness of monoclonal antibodies in the treatment of COVID-19 has been noticed [29,30,31]. South Africa is currently experiencing a surge of new COVID-19 cases driven by two Omicron sub-variants, BA.4 and BA.5 [32]. In this peculiar manner, the COVID-19 pandemic can potentially be self-fueling, with worldwide consequences.

Another challenge has been created by variants that emerged before the Omicron lineage was identified. Although the Delta variant, which led to a rapid de-escalation of previously dominant lineages in various world regions, was rapidly dominated by the Omicron variant, it was not entirely eliminated, and evidence suggests its cryptic circulation. A recent study based on wastewater-based epidemiology demonstrated that in Israel, the Delta variant continued to circulate under the rise of Omicron and was not entirely eliminated from the population. Based on the assumption of asymmetric cross-immunization in which protection from infection with the Delta variant in an individual previously infected with Omicron is four-fold lower than the protection from Omicron infection in a Delta-immunized person [33,34], it was modeled that the Omicron variant may eventually decrease its prevalence, while the Delta variant may maintain its circulation [35]. Although utilizing wastewater surveillance of SARS-CoV-2 has been previously evidenced to successfully identify shifts between viral variants accurately [36,37], it should be stressed that detecting cryptic variants may be challenging and should treated with caution as this technique does not yield a full viral genome and matching the sequenced parts to particular viral variants may be challenging. Nevertheless, if the prediction expressed by Yaniv et al. [35] comes true in regions such as Africa, this could lead to a novel wave of infections caused by Delta or its descendants and a possibly more significant burden to the healthcare system due to the increased clinical severity of this variant. However, ensuring a high vaccination coverage may prevent such a scenario.

Last but not least, the potential emergence of an entirely novel lineage of SARS-CoV-2 cannot be ignored. To this end, one should consider the following:(a)The receptor-binding domain (RBD) of the spike protein of the Omicron variant has no greater affinity to the angiotensin-converting enzyme 2 receptor than the RBD of the Delta variant, while some studies report it may even be lower [38]. Additionally, Omicron is less fusogenic than the Delta variant [39]. Furthermore, the viral loads in the upper airways also do not differ between these lineages [40]. The primary cause of the Omicron variant’s enhanced transmissibility is its ability to better evade the humoral immunity of vaccinated and convalescent individuals [41,42].(b)Research indicates that individuals infected with SARS-CoV-2 are the most contagious prior to symptom onset and during the symptomatic phase [43]. The time from symptom onset to death ranges in COVID-19 patients from 2 to 8 weeks (with a reported median of 16–19 days) due to acute respiratory distress symptoms, hyperinflammation, thrombosis, and other complications resulting from the exaggerated antiviral response [44,45]. Therefore, in the majority of cases, critically ill patients are not contagious, and their potential death has no consequences for viral fitness.

Therefore, a novel lineage of SARS-CoV-2 could potentially outcompete Omicron via an enhanced affinity to spike, increased furin and TMPRSS2 cleavage, and/or higher viral loads in the upper airways. This, in turn, may increase the odds of an overactive pro-inflammatory and cytotoxic immune response, ultimately translating into a greater clinical severity of COVID-19. In other words, an increase in the transmissibility of a SARS-CoV-2 variant does not necessarily have to be at the expense of clinical significance. The COVID-19 pandemic has already seen this being the case with the Delta lineage that, compared to preceding variants, was characterized by a higher basic reproduction number and induced higher viral loads, while its infections were significantly more severe.

## 3. Conclusions

In summary, the priority for COVID-19 vaccinations lies in preventing severe disease, hospitalizations, the need for mechanical ventilation, and death. The initial vaccination regime ensures this at a satisfactory level due to an adaptive cellular response, even if the serum level of neutralizing antibodies decreases over time [46,47,48,49,50]. Moreover, a booster dose with the original, Omicron-unoptimized vaccine is evidenced to enhance protection from the Omicron infection variant and further improve cellular immunity [51,52,53]. However, from the global perspective of pandemic control, priority must be given to vaccinating unvaccinated people in low-income countries and less-vaccinated populations due to high vaccine hesitancy [3,54,55]. This is despite the Omicron lineage having a lower clinical significance compared to other SARS-CoV-2 variants because it may: (i) continue to evolve with unpredicted consequences for COVID-19 severity, (ii) lose its fitness over a time due to high transmissibility and give rise to the reemergence of other variants such as Delta which could now be under cryptic circulation, and (iii) be outcompeted by entirely novel viral lineages with increased transmissibility and greater clinical severity. These issues must be taken into account when shaping recommendations and communication with the public in the low-income countries, which, contrary to many high-income regions, have sizable younger-age populations [56], a feature which lowers the percentage of severe infections and may additionally fuel vaccine hesitancy during circulation of the Omicron variant.

Better vaccine coverage in low-income countries can be achieved by direct dose donations and cross-subsidy, i.e., developed countries purchasing booster doses at higher prices to lower the cost of initial doses for low-income regions. The trading of COVID-19 vaccine doses between wealthy countries should no longer be tolerated. Simultaneously, significant educational efforts must be pursued in Africa to ensure vaccine uptake—this may be highly challenging due to high vaccine hesitancy and difficulties in understanding the long-term goals of COVID-19 vaccination when a less severe SARS-CoV-2 variant is circulating.

The pandemic is, by definition, a major epidemiological event of broad geographical spread. It has to be treated as such; otherwise, the global fight against SARS-CoV-2 will be prolonged, generating higher social and economic losses and costing more human lives.

## Figures and Tables

**Figure 1 vaccines-10-01070-f001:**
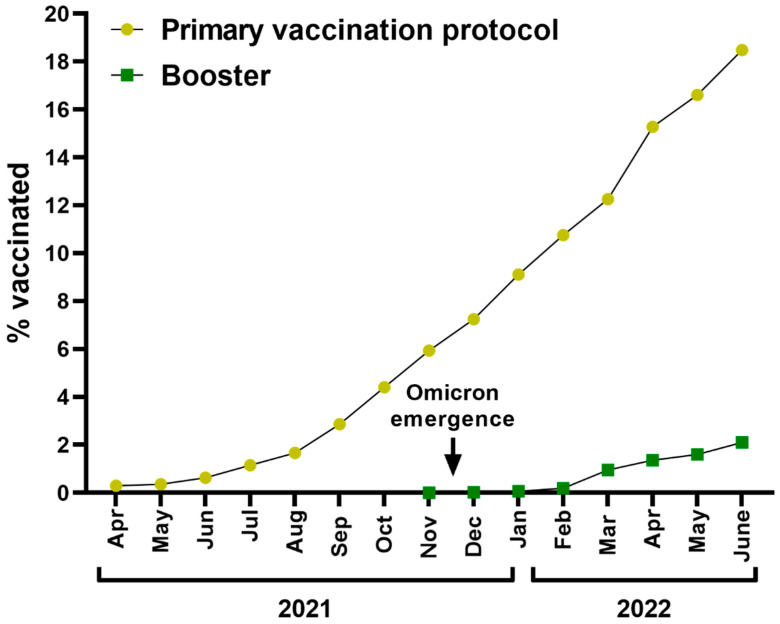
The percentage of the African population between April 2021 and June 2022 who completed the primary COVID-19 vaccination regime and were vaccinated with the booster dose (based on [8]).

## Data Availability

Not applicable.

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
