# Peer review of "The COVID-19 Vaccination Still Matters: Omicron Variant Is a Final Wake-Up Call for the Rich to Help the Poor"

_vaccines, 2022, doi:10.3390/vaccines10071070_

Round 1

Reviewer 1 Report

It’s a very interesting topic in this time when covid vaccination no longer seems a problem.

The Authors highlight  the fact that at the beginning of May 2022, only 16.6% of the African population had  completed an initiation vaccine protocol , compared to 73.6% in the European 48 Union and 66.4% in the USA. Generally, only 15.7% of people in low-income countries have received at least one dose.

I agree about the loss of interest in vaccination in regions such as Africa is built on the false premise that the Omicron is the final evolutionary step of SARS-CoV-2. The observation of milder clinical course of Omicron infections may lead to the assumption that high vaccine coverage is no longer needed, while the Omicron provides a final solution to the pandemic.

Another important point discussed is well established that SARS-CoV-2 can persist for months in patients with advanced HIV disease due to decreased immune response.

The pandemic control should find the solution to vaccinate unvaccinated people in low-income countries and less vaccinated populations due to high vaccine hesitancy in order to avoid the development of new variants of concern (VOC) . Infact Authors alert about the lack of attention in monitoring Omicron lineage because of the lower clinical significance compared to other SARS-CoV-2 variants  may: (i) continue to evolve with unpredicted consequences for COVID-19 severity, (ii) lose its fitness over a time due to high transmissibility and give rise to the reemergence of other variants such as Delta who are now under cryptic circulation, and (iii) be outcompeted by entirely novel viral lineage with increased transmissibility and greater clinical severity.

We hope Africa be helped to ensure vaccine uptake.

Author Response

Authors: Thank you very much for reviewing our manuscript and supporting the expressed assessment of the situation. We were glad to receive positive feedback on our submission.

Reviewer 2 Report

I think this is a very good article and the topic should be of interest to the Vaccines Journal readers. Even the manuscript is well written and well articulated, I think this could be improved following the below suggestions:

- Is unclear the type and number the COVID-19 vaccination needed in low-income countries. As an example, authors should be clear how many doses of COVID-19 vaccination is required (2 doses, 2 doses +1 booster, 2 doses + 2 boosters, etc), also the type population vaccinated is missing, is the recommendation of increasing COVID-19 vaccination should be only older adults, 18-64yrs, children... This is unclear right now from the manuscript, so probably the authors may provide more insights on what is their recommendation. Lastly, the overall number of doses required is also a big uncertainty, how many millions of doses should be sent to the low income countries (is there any estimation for this, any countries to prioritize), I think this is linked to my first suggestion on the vaccination scheme to be used. 

Also another point that was not mentioned is about how often those low-income countries should be updated with the Covid-19 boosters, we now know that covid vaccines lose their protection after a number of weeks, so an estimation on how many doses will be required to maintain a well protected population should also be considered, this is not only for later estimating the amount of doses needed to be produced for these regions but as well the economic distribution costs to be assumed by either donators or global organizations. The paper idea is a good one but requires some additional data in order that the message sounds compelling but at the same time feasible to achieve (from the epidemiological and economic perspective).

Author Response

Reviewer: I think this is a very good article and the topic should be of interest to the Vaccines Journal readers. Even the manuscript is well written and well articulated, I think this could be improved following the below suggestions.

Authors: Thank you very much for reviewing our manuscript and supporting the expressed assessment of the situation. We were glad to receive positive feedback on our submission and provide a revision following your recommendations.

Reviewer: Is unclear the type and number the COVID-19 vaccination needed in low-income countries. As an example, authors should be clear how many doses of COVID-19 vaccination is required (2 doses, 2 doses +1 booster, 2 doses + 2 boosters, etc), also the type population vaccinated is missing, is the recommendation of increasing COVID-19 vaccination should be only older adults, 18-64yrs, children... This is unclear right now from the manuscript, so probably the authors may provide more insights on what is their recommendation. Lastly, the overall number of doses required is also a big uncertainty, how many millions of doses should be sent to the low income countries (is there any estimation for this, any countries to prioritize), I think this is linked to my first suggestion on the vaccination scheme to be used. 

Authors: This is a very good point; thank you for raising it. We have added explained that:

"These regions should follow the recommendations on primary vaccination course, and administration of additional and booster doses in specific demographic groups, which are issued in high-income areas, e.g., the United States and the European Union. There is no indication that the low-income countries should have vaccine recommendations shaped differently to high-income regions because COVID-19 is a worldwide issue and should be treated equally regardless of one's origin or ethnicity."

We believe this makes our point very clear on how the vaccination recommendations should be shaped in LIC.

Reviewer: Also another point that was not mentioned is about how often those low-income countries should be updated with the Covid-19 boosters, we now know that covid vaccines lose their protection after a number of weeks, so an estimation on how many doses will be required to maintain a well protected population should also be considered, this is not only for later estimating the amount of doses needed to be produced for these regions but as well the economic distribution costs to be assumed by either donators or global organizations. The paper idea is a good one but requires some additional data in order that the message sounds compelling but at the same time feasible to achieve (from the epidemiological and economic perspective).

Authors: Thank you for this comment. However, at this point, we won't risk suggesting any threshold level for herd immunity as this concept is likely not achievable. What is achievable is slowing down the viral evolution with SARS-CoV-2 through vaccination. This is true that protection wanes 3-4 months from vaccination, but this mostly relates to protection from infection. In our manuscript, we indicate that the priority is to maintain good protection levels from severe disease, while this is achieved by cellular immunity, which is less prone to changes over time. At this point (low % of the population with a primary course of vaccination and even lower % of boosted individuals) in the low-income countries, we believe that vaccination efforts must be continued. Indicating a specific number of doses or coverage is highly challenging and outside the scope of our manuscript.

Reviewer 3 Report

Article that argues about the need for vaccination against SARS-CoV-2 in less developed countries where vaccination rates (and wealth) are low. I agree with most of the opinions raised by the authors.

The authors (and me too) disagree with the assumption that Omicron variant “may offer a natural solution to the pandemic”. However, this assumption is only clear in reference 15. In my opinion reference 16 only stated that Omicron infection indicates a “shorter period of illness and potentially of infectiousness which should impact work-health policies and public health advice.” But they don’t anticipate the end of the pandemic with the Omicron variant.

In fact, I think that most Governments of Europe, USA, Australia, etc. have down-scaled their policies in controlling SARS-CoV-2 pandemic on the basis of a lower hospital admission number with the current Omicron variants (let’s see what happens with BA.4 and BA.5 variants…) in addition to the high % of vaccinated population (at least with two and most people with three doses).

Besides, reference 15 is quite strange; it is only authored by one researcher that has many individual articles in the same Journal of which he was Editor-in-Chief.

Li-Wan-Po A. Journal of clinical pharmacy and therapeutics-Passing on the baton. J Clin Pharm Ther. 2022 Jan;47(1):2. doi: 10.1111/jcpt.13613. Epub 2022 Jan 23. PMID: 35066897.

Although it is not the objective of the article, it would have been also interesting to know the opinion of the authors about a fourth dose of the vaccine for vulnerable people in high-income countries as compared to non-vaccination in low-income countries.

Minor comments

The authors don´t comment on the population age on low-income countries, younger than in high-income countries. This would probably mean a lower percentage of severe infections and death caused by SARS-CoV-2 and more hesitancne to vaccination.

Lines 109-111: “A recent study based on wastewater-based epidemiology demonstrated that in Israel, the Delta variant continued to circulate under a rise of Omicron and was not entirely eliminated from the population.” Studies based on wastewater-based should be taken with caution. In our region, we genotype more than 80% of all cases (nasopharyngeal samples) and delta variant has totally disappear from patient circulation.

Line 171. Last word. Life or Lives?

Author Response

Reviewer: Article that argues about the need for vaccination against SARS-CoV-2 in less developed countries where vaccination rates (and wealth) are low. I agree with most of the opinions raised by the authors.

Authors: Thank you very much for reviewing our manuscript and supporting the expressed assessment of the situation. We were glad to receive positive feedback on our submission and provide a revision following your recommendations.

Reviewer: The authors (and me too) disagree with the assumption that Omicron variant "may offer a natural solution to the pandemic". However, this assumption is only clear in reference 15. In my opinion reference 16 only stated that Omicron infection indicates a "shorter period of illness and potentially of infectiousness which should impact work-health policies and public health advice." But they don't anticipate the end of the pandemic with the Omicron variant. In fact, I think that most Governments of Europe, USA, Australia, etc. have down-scaled their policies in controlling SARS-CoV-2 pandemic on the basis of a lower hospital admission number with the current Omicron variants (let's see what happens with BA.4 and BA.5 variants…) in addition to the high % of vaccinated population (at least with two and most people with three doses).

Authors: We agree with a Reviewer regarding the reference to the ZOE study. It was added to refer to this part of the sentence: "The observation of a milder clinical course of Omicron infections…." – we have moved it earlier in the sentence. Apart from the reference to Li-Wan-Po, we have added a reference to an article in Insider that presents contradictory views on Omicron and its potential role as an end game to the pandemic. It suits this part much better.

Reviewer: Besides, reference 15 is quite strange; it is only authored by one researcher that has many individual articles in the same Journal of which he was Editor-in-Chief. Li-Wan-Po A. Journal of clinical pharmacy and therapeutics-Passing on the baton. J Clin Pharm Ther. 2022 Jan;47(1):2. doi: 10.1111/jcpt.13613. Epub 2022 Jan 23. PMID: 35066897.

Authors: We wish to keep this citation because after all, it supports the referenced part. In addition, we have added one more reference, this time to the news article, that presents contradictory views expressed by different experts on the Omicron variant during the public conferences.

Reviewer: Although it is not the objective of the article, it would have been also interesting to know the opinion of the authors about a fourth dose of the vaccine for vulnerable people in high-income countries as compared to non-vaccination in low-income countries.

Authors: Thank you for this comment. Indeed, a second booster is not a subject of our manuscript, and evaluation of this strategy is currently subject to one of our other submission - currently subject to peer review. However, we agree that we should mention the authorization of the second booster and have added this passage in the revised manuscript:

"Generally, only 15.7% of people in low-income countries have received at least one dose [5]. At the same time, the high-income countries have increasingly authorized and recommended a second booster dose to maintain high levels of protection during upcoming waves of infections caused by the emerging Omicron sublineages [6,7]."

The references to the relevant EMA and FDA documents (ref. 6 & 7) were added.

Reviewer: The authors don't comment on the population age on low-income countries, younger than in high-income countries. This would probably mean a lower percentage of severe infections and death caused by SARS-CoV-2 and more hesitancne to vaccination.

Authors: This is a good point, and we have referenced it when discussing the points articulating why vaccination in low-income countries is still important. We have indicated that:

"These issues must be taken into account when shaping recommendations and communication with the public in the low-income countries, which, contrary to many high-income regions, have sizable younger-age populations [53], the feature which lowers the percentage of severe infections and may additionally fuel vaccine hesitancy during circulation of Omicron variant."

Reviewer: Lines 109-111: "A recent study based on wastewater-based epidemiology demonstrated that in Israel, the Delta variant continued to circulate under a rise of Omicron and was not entirely eliminated from the population." Studies based on wastewater-based should be taken with caution. In our region, we genotype more than 80% of all cases (nasopharyngeal samples) and delta variant has totally disappear from patient circulation.

Authors: Thank you for this comment. A Reviewer is right that wastewater monitoring has to be treated with caution. However, the study from Israel, published in Science of the Total Environment, is sound and delivered with no methodological issues. The same team has accurately detected a previous shift to the Alpha variant, and then to the Delta variant, by monitoring wastewater. Nevertheless, we agree that some caution must be emphasized, and this passage was added to the relevant part:

'Although utilizing wastewater surveillance of SARS-CoV-2 was previously evidenced to successfully identify shifts between viral variants [33,34] accurately, it should be stressed that detecting cryptic variants may be challenging and treated with caution as this technique does not yield full viral genome and matching the sequenced parts to particular viral variants may be challenging.'

Line 171. Last word. Life or Lives?

Authors: 'lives', of course. Thank you for noticing it. Corrected.

Reviewer 4 Report

I found this a very interesting and thought-provoking article.  Despite being 2 years into the pandemic there continues to be resurgence and emergence of new variants.  This paper summarises well the factors that can undermine the efforts to keep on top of infection rates, especially low rates of vaccination in resource limited countries. In a dynamic situation, the discussion is sufficiently up to date in terms of the current challenges.

I have no specific changes to recommend and just picked up typographical errors in Figure 1 'protocol' and on line 92 'of' not 'or'   

Author Response

years into the pandemic there continues to be resurgence and emergence of new variants.  This paper summarises well the factors that can undermine the efforts to keep on top of infection rates, especially low rates of vaccination in resource limited countries. In a dynamic situation, the discussion is sufficiently up to date in terms of the current challenges.

Authors: Thank you very much for reviewing our manuscript and supporting the expressed assessment of the situation. We were glad to receive positive feedback on our submission.

Reviewer: I have no specific changes to recommend and just picked up typographical errors in Figure 1 'protocol' and on line 92 'of' not 'or'   

Authors: The typo in the Figure has been corrected. Moreover, the data for June 2022 was added. The part in line 92 was corrected.